# Melatonin Maintains Postharvest Quality in Fresh *Gastrodia elata* Tuber by Regulating Antioxidant Ability and Phenylpropanoid and Energy Metabolism During Storage

**DOI:** 10.3390/ijms252111752

**Published:** 2024-11-01

**Authors:** Boyu Dong, Chengyue Kuang, Yulong Chen, Fangfang Da, Qiuping Yao, Dequan Zhu, Xiaochun Ding

**Affiliations:** 1School of Chinese Ethnic Medicine, Guizhou Minzu University, Guiyang 550025, China; kcy20000128@163.com (C.K.); 15772710928@163.com (Y.C.); dafangfang@gzmu.edu.cn (F.D.); wonderyqp@163.com (Q.Y.); zdqoliver@163.com (D.Z.); 2Key Laboratory of Guizhou Ethnic Medicine Resource Development and Utilization, Guizhou Minzu University, State Ethnic Affairs Commission, Guiyang 550025, China; 3Engineering Research Center for Fruit Crops of Guizhou Province, Key Laboratory of Plant Resource Conservation and Germplasm Innovation in Mountainous Region (Ministry of Education), College of Agriculture, Guizhou University, Guiyang 550025, China

**Keywords:** *Gastrodia elata*, melatonin, antioxidant capacity, phenylpropanoid metabolism, energy metabolism

## Abstract

Melatonin treatment has been reported to effectively preserve and improve the postharvest quality of fruits and vegetables during storage. This research focused on examining the significance of melatonin on maintaining the quality of fresh *Gastrodia elata* tubers throughout the storage period. The findings demonstrated that melatonin application effectively reduced the deterioration rate and inhibited the rise in respiratory rate, malondialdehyde content, and weight loss, while slowing down the decline in soluble solid content. Melatonin treatment led to a decrease in hydrogen peroxide production and a rise in non-enzymatic antioxidant concentrations, including ascorbic acid. Furthermore, it boosted both the activity and expression of indispensable antioxidant enzymes, like superoxide dismutase, catalase, and ascorbate peroxidase. Additionally, melatonin treatment promoted the accumulation of total phenols, flavonoids, and lignin in fresh *G. elata*, while enhancing both the activity and expression of critical enzymes in the phenylpropanoid pathway, including phenylalanine ammonia-lyase, cinnamate-4-hydroxylase, and 4-coumarate-CoA ligase. Moreover, melatonin treatment boosted the activity and expression of energy-associated enzymes including H^+^-ATPase, succinate dehydrogenase, Ca^2+^-ATPase, and cytochrome C oxidase, contributing to the improvement of energy levels in fresh *G. elata*. In summary, melatonin enhances the antioxidant potential and reduces oxidative damage in fresh *G. elata* by activating reactive oxygen species, phenylpropanoid metabolism, and energy metabolism, thereby maintaining its postharvest quality.

## 1. Introduction

*Gastrodia elata*, classified in the genus *Gastrodia* in the *Orchidaceae* family, is a tuberous plant known for its dual role as both a food and medicinal resource [1]. After harvesting, it is typically processed through drying and sulfur fumigation to preserve its quality [2]. However, these methods have the following drawbacks: (1) Loss of nutritional and bioactive components. (2) Moisture loss, leading to a poor texture. With the growing recognition of its nutritional value, the market demand for fresh *G. elata* has significantly increased. However, fresh *G. elata* is highly perishable and prone to spoilage, and there is currently a lack of systematic research on its storage.

Melatonin, a compound derived from the essential amino acid tryptophan, is widely recognized as a novel plant hormone due to its diverse functions [3]. It is generally regarded as safe, with studies indicating that it is well-tolerated even at high doses. The U.S. Food and Drug Administration (FDA) classifies melatonin as low-risk, with only mild side effects such as drowsiness. In China, melatonin is legally approved for health products under regulations that ensure proper dosage and labeling, as indicated by the National Medical Products Administration (NMPA). Prior studies have demonstrated that melatonin can enhance the quality of fruits and vegetables by improving antioxidant capacity, maintaining energy levels, and activating phenylpropanoid metabolism. Postharvest treatments with melatonin have been demonstrated to maintain storage quality and postpone the deterioration of passion fruit [4], green pepper [5], white mushroom [6], and strawberry [7]. Nonetheless, the influence of melatonin application on the postharvest attributes of fresh *G. elata* is still not well understood.

Studies have confirmed that reactive oxygen species (ROS) serve a dual function in the defensive responses of plants, both by directly attacking pathogens and functioning as messenger molecules that regulate defense gene expression. Conversely, an overproduction of ROS can harm cellular structures, weaken membrane integrity, and hasten the aging process in plants [8]. Plants maintain the ROS balance by utilizing antioxidant enzymes like superoxide dismutase (SOD), ascorbate peroxidase (APX), and catalase (CAT) [9]. Previous research has also demonstrated that postharvest treatments like melatonin [4], Acibenzolar-S-methyl [10], and hydrogen-rich water [11] can amplify the antioxidant potential of fruits and vegetables.

Phenylpropanoid metabolism is a unique secondary metabolic pathway in plants, having a significant impact in their antioxidant and disease resistance processes. Many key antimicrobial compounds in plants, including phenolic, flavonoids, and lignin, are synthesized through phenylpropanoid metabolism [12]. Additionally, the accumulation of phenylpropanoid metabolites can enhance the antioxidant capacity of mango [13], pear [14], and apple fruits [15].

The energy supply within cells is crucial for conserving the postharvest quality of fruits and vegetables. The senescence and quality deterioration of plants after harvest are closely related to energy processes [16]. Prior studies have confirmed that different postharvest treatments can enhance the energy levels in fruits and vegetables, thereby stabilizing internal metabolism and delaying the aging process [17,18,19]. However, the effects of melatonin on postharvest quality in fresh *G. elata* remain unknown.

This research seeks to investigate the significance of melatonin treatment on the quality, antioxidant capacity, and phenylpropanoid and energy metabolism of fresh *G. elata*. By gaining a deeper understanding of how melatonin helps maintain the quality of fresh *G. elata*, the research will contribute to the establishment of effective and user-friendly methods for postharvest storage and preservation.

## 2. Results

### 2.1. Influence of Melatonin on Spoilage, Respiratory Rate, Malondialdehyde (MDA), Soluble Solid Content (SSC), and Weight Loss in G. elata Tubers

According to Figure 1A, during the first 6 days, none of the groups, including both the different melatonin treatment groups and the control group, exhibited tubers spoilage. Compared to the control group, the spoilage rate in the 50 μmol/L melatonin-treated group was markedly less on days 8 and 10, and it showed better spoilage inhibition compared to the 25 and 100 μmol/L melatonin treatment groups.

As seen in Figure 1B,D, the fresh *G. elata* in the 50 μmol/L melatonin treatment group had consistently lower respiration rates and MDA content for the entire storage interval, remarkably outperforming the control and other melatonin concentrations. Based on the analysis of spoilage rates, 50 μmol/L was identified as the optimal melatonin concentration for further experiments.

Both the melatonin treatment and the control group displayed a rising trend in weight loss, but on days 4 and 8 after treatment, the weight loss rate of fresh *G. elata* in the melatonin treatment group showed a significantly reduced level in comparison to the control (Figure 1C).

Melatonin treatment resulted in significantly lower SSC on days 6 to 10 in fresh *G. elata* compared with the control (Figure 1E).

### 2.2. Effect of Melatonin on ROS Metabolism

Throughout the entire storage period, the hydrogen peroxide (H_2_O_2_) content in the melatonin-treated fresh *G. elata* tubers consistently decreased compared to the control group. Notably, significant differences were observed from days 6 to 10 (Figure 2A).

In the melatonin treatment group, the ascorbic acid (AsA) content increased from days 0 to 4 and then decreased. In contrast, the AsA content fluctuated from days 0 to 4 before starting to decline in the control. Overall, the AsA content in the melatonin treatment group remained obviously elevated across the storage duration (Figure 2B).

In the melatonin treatment and the control group, the SOD activity increased from days 0 to 4, followed by an overall decline. However, from days 6 to 10, the SOD activity in the melatonin treatment of fresh *G. elata* was notably enhanced in comparison with the control (Figure 2C). According to Figure 2D, melatonin treatment distinctly increased the expression of the *GeSOD* from days 4 to 10.

The CAT activity in both the melatonin treatment and control groups presented a comparable trend throughout the storage duration, increasing from days 0 to 8 and then decreasing. Obviously, the melatonin treatment group had remarkably greater CAT activity than the control from days 6 to 10 (Figure 2E). The trend in *GeCAT* expression mirrored the enzyme activity, with the melatonin treatment group showing substantially greater expression from days 6 to 10, while differences were not significant at other times (Figure 2F).

Regarding the APX activity, in both the melatonin treatment and the control groups, fresh *G. elata* increased from days 0 to 4, after which the that in control rapidly declined. The APX activity in melatonin-treated fresh *G. elata* decreased from days 4 to 8 but then slightly increased, remaining appreciably enhanced from days 6 to 10 (Figure 2G). After melatonin treatment, the expression of the *GeAPX* in fresh *G. elata* was substantially greater in the melatonin treatment group from days 4 to 10 compared to the control (Figure 2H).

### 2.3. Measure the Levels of Adenosine Triphosphate (ATP), Adenosine Diphosphate (ADP), Adenosine Monophosphate (AMP), and Energy Charge

In the melatonin treatment group, the ATP content increased from days 0 to 2 and then reduced, while the levels in the control group consistently declined. On days 2 to 6 and 10, the ATP levels in the melatonin treatment group were considerably greater than those of the control (Figure 3A).

The ADP content decreased continuously in both the melatonin treatment group and the control, but the melatonin treatment group had markedly higher ADP levels from days 2 to 8 compared to the control (Figure 3B).

According to Figure 3C, the AMP content in the melatonin-treated fresh *G. elata* decreased consistently during the storage period, with particularly significant differences from days 4 to 10.

The energy charge in the melatonin treated group was appreciably enhanced in comparison to the control during the storage period from days 4 to 10 (Figure 3D).

### 2.4. Assessment of Energy Metabolism Enzyme Activities

Throughout the entire experiment, both the control and melatonin treatment group presented a declining trend in H^+^–ATPase activity. However, melatonin treatment slowed this decline. From days 4 to 8, the melatonin-treated group exhibited a marked rise in H^+^–ATPase activity relative to the control (Figure 4A). Additionally, the melatonin group displayed a steady increase in *GeH^+^*–*ATPase* expression when compared to the control group across the whole preservation time (days 2 to 10) (Figure 4B).

In the melatonin treatment group, Ca^2+^-ATPase activity increased from days 0 to 4, then fluctuated from days 4 to 8, and subsequently decreased. A significant elevation in Ca^2+^-ATPase activity was observed in the melatonin group, relative to the control, on days 2–4 and 8–10 (Figure 4C). Following melatonin treatment, the expression of *GeCa^2+^*-*ATPase* in fresh *G. elata* appreciably increased from days 2 to 8 (Figure 4D).

In the melatonin treatment group, the succinate dehydrogenase (SDH) activity improved from days 0 to 2 and reached its peak, and then fluctuated. In contrast, SDH activity in the control group dropped continuously from days 0 to 8 before rising again. During the period from days 6 to 8, the SDH activity in the melatonin treatment group was considerably greater compared with the control (Figure 4E), and the expression of the *GeSDH* in the treatment group was notably heightened during days 2 to 4 and 8 to 10 (Figure 4F).

After melatonin treatment, cytochrome C oxidase (CCO) activity continuously increased and reached its peak from days 0 to 4, then decreased, with remarkably greater activity on days 4 to 6 and 10 as opposed to the control (Figure 4G). According to Figure 4H, the melatonin treatment group exhibited a sustained increase in *GeCCO* expression during the entire storage duration.

### 2.5. Effects of Melatonin Treatment on Total Phenolic, Flavonoid, and Lignin Contents

Over the duration of the experiment, the melatonin-treated group had a substantially higher total phenol content from days 2 to 8 (Figure 5A).

Throughout the experiment, flavonoid content in the melatonin-treated group remained consistently higher than in the control, with the most pronounced differences appearing in the later stages (days 8 to 10) (Figure 5B).

The lignin content in both the control and melatonin treatment groups followed a similar trend, peaking on day 4 and then fluctuating. However, between days 4 and 10, the lignin levels in the melatonin treatment group were notably elevated compared with the control (Figure 5C).

### 2.6. Effect of Melatonin on Phenylpropanoid Pathway

After melatonin treatment, the phenylalanine ammonia-lyase (PAL) activity in fresh *G. elata* remained substantially increased compared to the control group over the full course of storage, with particularly significant differences from days 4 to 8 (Figure 5D). The *GePAL* expression in the treatment group was also considerably enhanced compared to the control (Figure 5E).

No notable difference in cinnamate-4-hydroxylase (C4H) activity was detected between the control and melatonin-treated groups during days 0 to 4. However, from days 6 to 10, the melatonin group showed a pronounced increase in C4H activity compared to the control (Figure 5F). Furthermore, the *GeC4H* expression in the melatonin group was markedly higher from days 4 to 10 (Figure 5G).

The 4-coumarate-CoA ligase (4CL) activity in the melatonin treatment group increased rapidly from days 0 to 6, then decreased, but remained markedly enhanced compared to the control during the later stages of the experiment (days 6 to 10) (Figure 5H). With the exception of day 4, the *Ge4CL* expression in the melatonin treatment group was appreciably greater in comparison to the control at all other time points (Figure 5I).

## 3. Discussion

After harvest, swift respiration and transpiration rates act as a crucial function in the aging and degradation of fruits and vegetables. Common preservation methods aim to reduce respiration intensity and transpiration [20]. This research confirms that, compared to other melatonin concentrations, treatment with 50 μmol/L melatonin most effectively inhibits postharvest spoilage in fresh *G. elata* while significantly reducing respiration and weight loss rates. This implies that melatonin treatment can adequately slow down the aging and spoilage process of fresh *G. elata*, helping to maintain postharvest quality. Prior investigations have also found that melatonin treatment markedly slows the rise in respiration and weight loss rates during storage in Chinese flowering cabbage [21]. Additionally, apple fruits treated with melatonin showed appreciably reduced spoilage and weight loss rates when contrasted with the control [22].

MDA, a byproduct of lipid peroxidation, is commonly utilized as a key marker for fruit aging. This investigation revealed that melatonin treatment significantly lowers MDA levels in fresh *G. elata*, suggesting that postharvest melatonin treatment helps maintain the integrity of its cell membranes. Comparable findings have been observed in research on melatonin-treated apple [22] and litchi fruits [23]. During postharvest respiration, SSC is gradually consumed as a substrate. However, excessive respiration rates can accelerate SSC depletion [24]. The results demonstrate that melatonin treatment successfully minimizes SSC consumption in fresh *G. elata* while in storage. Earlier research has also confirmed that melatonin treatment helps maintain SSC levels in orange fruits [25]. Similarly, studies on sweet cherry have yielded consistent results [26].

Plants rapidly generate significant amounts of ROS in response to biotic or abiotic stimuli. In small concentrations, ROS can act as signaling agents to trigger the expression of genes involved in defense mechanisms [27]. On the other hand, high levels of ROS may cause damage to plants. Antioxidant enzymes within the plant help to detoxify excess ROS, maintaining ROS homeostasis [28]. SOD is one of the primary sources of H_2_O_2_ within cells. It converts unstable superoxide radicals into the relatively stable H_2_O_2_, thereby reducing oxidative damage to the plant. Subsequently, APX and CAT can break down H_2_O_2_ into H_2_O and O_2_. Additionally, the non-enzymatic antioxidant AsA can also help remove excess H_2_O_2_ [20]. This investigation demonstrates that melatonin treatment can inhibit the increase in H_2_O_2_ levels in fresh *G. elata* while in storage, while enhancing the activities of SOD, APX, and CAT, as well as the content of AsA. These findings suggest that postharvest melatonin treatment activates antioxidant enzyme activity and increases non-enzymatic antioxidant levels, thereby maintaining stable ROS levels in fresh *G. elata* and preventing oxidative damage. Cai et al. showed that after melatonin treatment, the activities of SOD and CAT, as well as the AsA content in passion fruit, were substantially boosted compared with the control [4]. Similarly, Wang et al. found that melatonin treatment effectively reduced H_2_O_2_ levels in postharvest melon, while raising the activities of SOD, CAT, and APX, together with increasing AsA content, thus maintaining fruit quality [29]. Additionally, corresponding outcomes were observed in research on melatonin-treated mangoes [30]. The discoveries presented suggest that melatonin treatment has the possibility to promote both the activity and gene expression of antioxidant-associated enzymes, including SOD, CAT, and APX, while simultaneously increasing the concentration of non-enzymatic antioxidant AsA, thereby maintaining the ROS balance in fresh *G. elata*.

Phenylpropanoid metabolism is one of the key pathways in plant secondary metabolism, responsible for the biosynthesis of phenolic compounds. The enzymes PAL, 4CL, and C4H are crucial in regulating this pathway [31]. PAL functions as the foremost enzyme, converting L-phenylalanine into cinnamic acid. Subsequently, C4H and 4CL act on various hydroxycinnamic acids, converting them into thioesters that contribute to the production of phenolic compounds, flavonoids, and anthocyanins [32]. Phenolic compounds, which serve as precursors for lignin and other disease-resistant substances, possess antimicrobial properties and promote lignin accumulation in the cell wall [33]. Our analysis confirmed that melatonin treatment markedly enhanced the activities and gene expression of PAL, C4H, and 4CL in fresh *G. elata* tubers over the storage period, while also encouraging the accumulation of total phenols, flavonoids, and lignin. Earlier findings suggest that melatonin treatment can significantly elevate PAL activity in pomegranate fruits, leading to a much higher accumulation of total phenols compared to the control group [34]. Furthermore, studies have shown that treating guava fruits with exogenous melatonin increases their total phenolic content, thereby preserving antioxidant capacity and improving their resistance [35]. Similar findings have been documented in studies involving melatonin treatment of papaya [36] and sweet cherry [37]. Therefore, we hypothesize that melatonin treatment can activate the phenylpropanoid pathway, leading to the rapid gathering of total phenols, flavonoids, and lignin. This, in turn, enhances the antioxidant capacity of fresh *G. elata* and helps maintain its quality during storage.

ATP is typically regarded as the primary energy source for life processes, and its deficiency can disrupt plant metabolism, leading to cell membrane damage and apoptosis under stress conditions [38]. The energy generated through metabolic processes is closely linked to the aging and quality of postharvest fruits [39]. This investigation demonstrated that melatonin treatment can enhance energy levels in postharvest *G. elata*. Tan et al. identified that exogenous melatonin treatment boosted ATP levels by enhancing mitochondrial energy metabolism, which helped maintain the quality characteristics of Chinese flowering cabbage during storage [21]. Similarly, melatonin treatment maintained higher intracellular energy levels in white mushrooms, effectively delaying senescence and preserving their storage quality [6]. Further studies have revealed that maintaining high energy levels through treatments with oxalic acid [40], exogenous ATP [41], and 1-methylcyclopropene [42] can effectively improve postharvest quality in peaches and Nanguo pears.

The regulation of boosted energy levels is significantly linked to the function of multiple mitochondrial enzymes. H^+^-ATPase, Ca^2+^-ATPase, SDH, and CCO serve vital functions in facilitating oxidative phosphorylation and ATP production in plant systems [40]. This investigation revealed that the treatment with melatonin greatly enhances both the activity and gene expression of H^+^-ATPase, Ca^2+^-ATPase, SDH, and CCO in fresh *G. elata*. Following melatonin treatment, the activity of H^+^-ATPase, Ca^2+^-ATPase, SDH, and CCO was notably increased in postharvest lichi fruits [23,43]. Similarly, research on melatonin-treated lotus seeds [44] and tomatoes [45] yielded comparable results. These results suggest that melatonin treatment can enhance the energy levels of postharvest fresh *G. elata*. Adequate energy supply helps maintain the regular activity of the metabolic system in fresh *G. elata*, slowing down the aging process during storage and preserving its quality as much as possible. 

These antioxidant, phenylpropanoid, and energy pathways are interconnected and mutually supportive. ATP generated from enhanced energy metabolism supports the biosynthesis of phenolic compounds and antioxidant enzymes [43]. In turn, antioxidants maintain mitochondrial integrity, preventing oxidative stress that could impair ATP production [23,43]. The accumulation of lignin and other phenolic compounds further reinforces the cell wall, reducing respiration and microbial infection [13,33]. This circular relationship ensures that *G. elata* maintains both physiological stability and nutritional quality during prolonged storage.

In summary, the ability of melatonin to activate multiple physiological pathways—each essential for maintaining quality—highlights its potential as a powerful postharvest treatment. By regulating ROS metabolism, enhancing phenylpropanoid biosynthesis, and boosting energy production, melatonin ensures that the postharvest deterioration of *G. elata* is minimized, enabling longer storage while maintaining nutritional and sensory qualities. These findings provide new insights into the broad physiological relevance of melatonin, not only in *G. elata* but also across other horticultural products, reinforcing the importance of targeted postharvest interventions.

## 4. Materials and Methods

### 4.1. G. elata and Chemicals

*G. elata* tubers was collected in early October, just before dormancy, on a clear morning in Zhaotong City, Yunnan Province, China (longitude 103°79′ E, latitude 27°38′ N). After collection, the *G. elata* tubers was transported to the laboratory, where undamaged samples, free from insect infestations and mechanical injuries, with yellowish-white stems and intact arrow-shaped buds, were carefully selected. Solarbio provided support for melatonin.

### 4.2. Treatment

After removing soil by rinsing *G. elata* tubers with running water, the samples were sterilized on the surface by immersing in 1% sodium hypochlorite for 3 min, followed by three rinses with distilled water. The *G. elata* tubers were then immersed in melatonin solutions at concentrations of 25, 50, and 100 μM for 10 min, with distilled water serving as a control. Post-treatment, the *G. elata* tubers were air-dried at room temperature in a ventilated area, placed in trays, and stored at 50–60% relative humidity and 22 ± 1 °C. Each treatment was replicated three times, with 120 *G. elata* samples per replication.

### 4.3. G. elata Tubers Collection

*G. elata* tubers were stored at ambient temperature, with samples collected after 0, 2, 4, 6, 8, and 10 days. Tissue samples were peeled from the equatorial region, approximately 3–6 mm thick. The samples were then immediately frozen in liquid nitrogen and stored at −80 °C.

### 4.4. Determination of Respiratory Rate, MDA, Weight Loss, Deterioration Rate, and SSC

Respiration rate is measured using a portable gas analyzer HM-GX10 (Shandong Hengmei Electronic Technology Co., Ltd., Weifang, China). The *G. elata* tubers were placed in preservation bags and left to rest for 40 min before measuring CO_2_ concentration on days 0, 2, 4, 6, 8, and 10. Respiration rate is expressed as CO_2_ mL/kg/h. Each treatment is repeated three times, using 8 samples per replicate.

The MDA content was measured following the procedure outlined by Jincy et al. [46]. Absorbance was recorded at wavelengths of 450 nm, 532 nm, and 600 nm, and the MDA levels were expressed in mmol kg^−1^ of fresh weight (FW).

To determine the deterioration rate, three replicates are performed for each treatment, with 100 samples of fresh *G. elata* tubers collected in each replicate. The deterioration rate (%) was determined by applying the following formula:$$Deterioration\;rate\left(\%\right)=\frac{\mathrm{Number}\;\mathrm{of}\;\mathrm{decayed}\;\mathrm{samples}}{\mathrm{Total}\;\mathrm{number}\;\mathrm{of}\;\mathrm{samples}}\;\times \;100$$
$$Weight\;loss\;rate\left(\%\right)=\frac{\mathrm{Weight}\;\mathrm{before}\;\mathrm{storage}-\mathrm{weight}\;\mathrm{after}\;\mathrm{storage}}{Weightbeforestorage}\;\times \;100$$

Fresh *G. elata* samples were pressed and filtered through four layers of gauze to measure SSC. by using a Brix refractometer (ATAGO Co., Ltd., Tokyo, Japan), and the results given as a percentage.

### 4.5. Determination of H_2_O_2_ and AsA Content and Antioxidant-Related Enzyme Activity

To quantify the H_2_O_2_ content in *G. elata*, a modified extraction and measurement protocol was adapted from Dong et al. [33]. Initially, a 3 g sample of frozen tissue was weighed and placed in a pre-cooled homogenization container. Subsequently, 3 mL of cold acetone, kept at 4 °C, was added to the sample. The mixture was homogenized thoroughly using a homogenizer. Subsequently, the homogenate was subjected to centrifugation at 9000× *g* for 20 min at 4 °C. Following centrifugation, the supernatant was carefully collected, avoiding the disturbance of the pellet, and 1 mL of the supernatant was aliquoted into a clean tube. To initiate the reaction for H_2_O_2_ determination, 200 μL of 20% titanium tetrachloride solution was added to the supernatant. Following the washing steps, the precipitate was redissolved in 3 mL of 1 M sulfuric acid (H_2_SO_4_). After the dissolution, the sample was centrifuged again at 9000× *g* for 10 min at 4 °C to clarify the solution. The absorbance of the resulting supernatant was measured at a wavelength of 410 nm using a spectrophotometer. The H_2_O_2_ concentration was subsequently determined in millimoles per kilogram of FW.

The quantification of AsA in the *G. elata* tissue was carried out utilizing an assay kit obtained from the Nanjing Jiancheng Bioengineering Institute (Nanjing, China). 1 g of fresh tissue was weighed and promptly homogenized in 6% (*w/v*) metaphosphoric acid (HPO_3_) solution. The homogenate was then centrifuged at 10,000× *g* for 10 min at 4 °C. For the assay, 200 μL of reagent was added to the supernatant, followed by a brief vortexing. The reaction mixture was incubated for 30 min at room temperature. After incubation, the absorbance was measured at 534 nm using a spectrophotometer. The AsA concentration was calculated using the standard curve ranging from 0.1 g/L to 1 g/L, and the final values were expressed in g kg^−1^ FW.

The activities of SOD, CAT, and APX were measured following the methods described by Dong et al. [33]. To determine the SOD activity, a reaction mixture was prepared by combining 0.4 mL of the crude enzyme extract with 1.5 mL of 50 mM sodium phosphate buffer (pH 7.8), 0.3 mL of 140 mM methionine, 0.3 mL of 100 μM ethylenediaminetetraacetic acid (EDTA), 0.3 mL of 740 μM nitro blue tetrazolium (NBT), and 0.3 mL of 20 μM riboflavin. The formation of blue formazan was monitored by measuring absorbance at 560 nm, with enzyme activity expressed as 103 U·kg^−1^ protein, where one unit (U) is defined as the amount of enzyme causing 50% of NBT reduction.

For measuring the CAT activity, the reaction was initiated with 3 mL of 0.01 M H_2_O_2_ and 0.3 mL of crude enzyme extract by monitoring absorbance changes at 240 nm. CAT activity was reported as 10^3^ U kg^−1^ protein, with 1 U defined as the amount of enzyme causing a 0.01 change in absorbance at 240 nm per minute.

The APX activity was determined using 2 mL of 100 mM sodium phosphate buffer (pH 7.6), 0.2 mL of enzyme extract, and 0.5 mL of 30% (*v/v*) H_2_O_2_ in the reaction mixture. Enzyme activity was reported as 10^3^ U·kg^−1^ protein, where 1 U was defined as the amount of enzyme that produces a 0.01 change in absorbance at 340 nm per minute.

### 4.6. Determine the Content of AMP, ADP, and ATP, Energy Charge, and Energy Metabolism-Related Enzyme Activity Determination

The concentrations of AMP, ADP, and ATP were extracted and measured in accordance with the method outlined by Shu et al. [19]. A 3 g frozen tissue sample was mixed with 5 mL of 0.6 M perchloric acid, and the mixture was centrifuged at 11,000× *g* for 30 min at 4 °C. The resulting supernatant was neutralized to approximately pH 6.8 with 1 M potassium hydroxide (KOH), diluted to a final volume of 5 mL, and filtered through a 0.45 µm membrane. High-performance liquid chromatography (HPLC) was used to quantify ATP, ADP, and AMP concentrations, with the master standard prepared following Shu et al. [19]. The concentrations were reported in g kg^−1^. The energy charge was determined using the following formula: (1/2 ADP + ATP)/(AMP + ADP + ATP). 

Additionally, the activities of H^+^-ATPase, Ca^2+^-ATPase, CCO, and SDH were determined using commercial assay kits from Jiancheng Bioengineering (Nanjing, China), following the manufacturer’s instructions. H^+^-ATPase and Ca^2+^-ATPase activities were determined by measuring the release of phosphorus, initiated by adding 100 µL of 0.03 mol L^−1^ ATP-Tris (pH 8.0) and halted with 5% (*w/v*) trichloroacetic acid (TCA) after 20 min of incubation at 37 °C. One unit of activity was defined as the release of 1 μmol of phosphorus per minute, measured as absorbance at 660 nm. SDH activity was using an assay medium that contained 0.3 mL of crude mitochondria extract, 3 mL of 0.2 mmol L^−1^ potassium phosphate buffer (pH 7.4), 1 mL of 0.2 mmol L^−1^ sodium succinate, 0.1 mL of 1 mmol L^−1^ di-p-chlorophenylmethyl carbinol, and 0.1 mL of 0.33% (*w/v*) methyl sulfenyl phenazine. One unit of SDH activity was defined as an increase of 0.01 in absorbance per minute at 600 nm. CCO activity was assayed using 0.2 mL of crude mitochondria extract, 0.2 mL of 0.04% (*w/v*) cytochrome c solution, and 0.5 mL of 0.4% (*w/v*) dimethyl phenylene diamine. One unit of CCO activity was defined as an increase of 0.1 in absorbance per minute at 510 nm [40]. 

### 4.7. Analysis of Enzyme Activities and Metabolite Contents Related to the Phenylpropanoid Pathway

Total phenolics, flavonoids, and lignin were obtained following the procedure outlined by Liu et al. [47]. The concentrations of total phenolics and lignin are reported as ΔOD280·g^−1^ FW, while flavonoid content is measured as ΔOD325·g^−1^ FW.

The reaction system for the PAL activity included 5 mL of 0.05 M sodium borate buffer (pH 8.7), supplemented with 5 mM mercaptoethanol, 2 mM EDTA, and 18 g L^−1^ polyvinylpolypyrrolidone (PVPP). The PAL activity was determined following the method of Li et al. [48]. A 1 mL aliquot of crude enzyme extract was incubated with 3 mL of L-phenylalanine at 37 °C for 60 min, and the absorbance of the reaction was recorded at 290 nm. The PAL activity was expressed as U·mg^−1^ protein, where 1 U was defined as a 0.01 increase in absorbance at 290 nm per minute.

For measuring the C4H activity, a solution was prepared by mixing 5 mL of 50 mM Tris–HCl buffer (pH 8.8) with 4.5 mM magnesium chloride, 14 mM mercaptoethanol, 8 µM rifampicin, 1.2 mM phenylmethylsulfonyl fluoride (PMSF), 0.9 g L^−1^ PVPP, 4 mM ascorbic acid (AsA), and 9% (*v/v*) glycerol. The C4H activity was also measured using the method of Li et al. [48] and expressed as 0.01 ΔOD340·mg^−1^ protein.

The reaction system for 4CL consisted of 0.2 M Tris–HCl buffer (pH 7.6), 20% glycerol, and 0.09 M dithiothreitol (DTT). Similarly, the 4CL activity was analyzed according to Li et al. [48]. The reaction mixture contained 0.6 mL of crude enzyme, 0.50 mL of 75 mM magnesium chloride, 0.15 mL of 1 M coenzyme A, 0.15 mL of 2 mM p-coumaric acid, and 0.15 mL of 0.8 mM ATP. Activity was reported as U·mg^−1^ protein, where 1 U corresponds to a 0.01 ΔOD333 increase per minute.

### 4.8. RNA Extraction and First-Strand cDNA Synthesis

Total RNA was isolated from 1.0 g of *G. elata* using the RNeasy Plant Mini Kit (Takara, Kyoto, Japan), following the manufacturer’s instructions precisely. First-strand cDNA was synthesized from 1 µg of total RNA using the Reverse-iT™ 1st Strand Synthesis Kit (Takara, Japan), adhering strictly to the provided protocols.

### 4.9. Real-Time Quantitative (RT-PCR)

For RT-PCR, the reaction mixture included 1 µL of template cDNA, 1 µL of 10 µM forward and reverse primers, and 10 µL of SYBR™ Green qPCR Master Mix in each well, with a total reaction volume of 20 µL. The ABI PRISM™ 7000 Sequence Detection System (Applied Biosystems, Waltham, MA, USA) was used for analysis, and *Actin* was utilized as the reference gene. Each sample was analyzed in triplicate, and the specific primers used are listed in Appendix A. The relative gene expression was quantified against the *Actin* control using the 2^−ΔΔCt^ method.

### 4.10. Data Analysis

Data analysis was conducted using the SPSS software program version 19.0. A one-way analysis of variance (ANOVA) was applied, with significant differences determined at *p* < 0.05. For pairwise comparisons between control and melatonin treatment, a Student’s *t*-test was applied.

## 5. Conclusions

This research demonstrates that melatonin treatment significantly enhances the quality of fresh *G. elata* tubers during storage by reducing the deterioration rate, respiratory rate, malondialdehyde content, and weight loss, thus preserving postharvest quality. Melatonin also boosts antioxidant capacity by increasing the activity and expression of key enzymes such as SOD, CAT, and APX, minimizing oxidative damage. In terms of phenylpropanoid metabolism, melatonin promotes the accumulation of phenolic compounds, flavonoids, and lignin, enhancing the activity of critical enzymes like PAL and 4CH. Additionally, melatonin treatment enhances energy metabolism by stimulating enzymes such as H^+^-ATPase and CCO, improving energy levels and delaying cell membrane damage. In summary, melatonin effectively improves the quality, antioxidant capacity, phenylpropanoid metabolism, and energy metabolism of fresh *G. elata*, preserving its postharvest quality. These findings underscore the potential of melatonin as a natural agent for enhancing the postharvest management of *G. elata*, contributing to an improved market value for this valuable tuber.

## Figures and Tables

**Figure 1 ijms-25-11752-f001:**
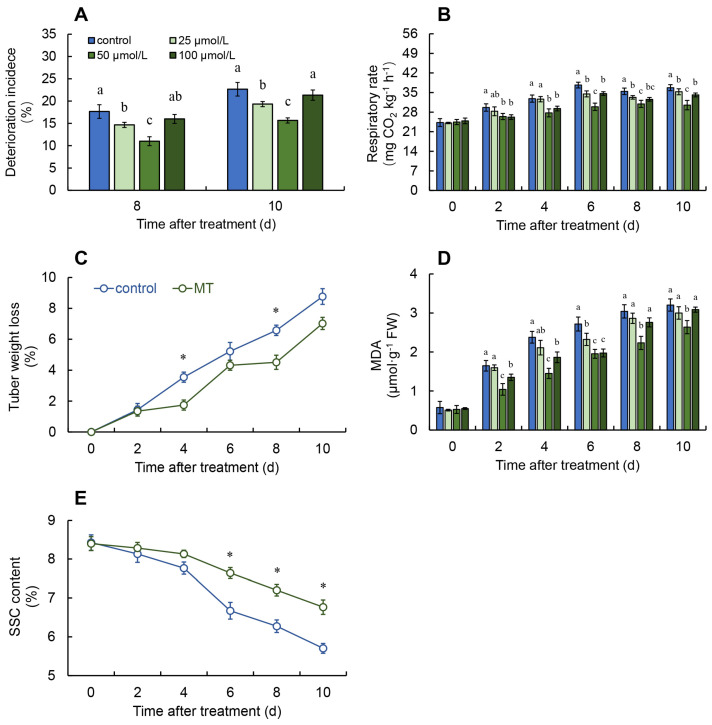
Effect of melatonin treatment on the deterioration incidence (**A**), respiratory rate (**B**), malondialdehyde (MDA) (**C**), weight loss (**D**), and soluble solid content (SSC) (**E**) in fresh *G. elata*. * and different letters indicate statistically significant differences (*p* ≤ 0.05). Vertical bars represent the standard errors of the means (±SE).

**Figure 2 ijms-25-11752-f002:**
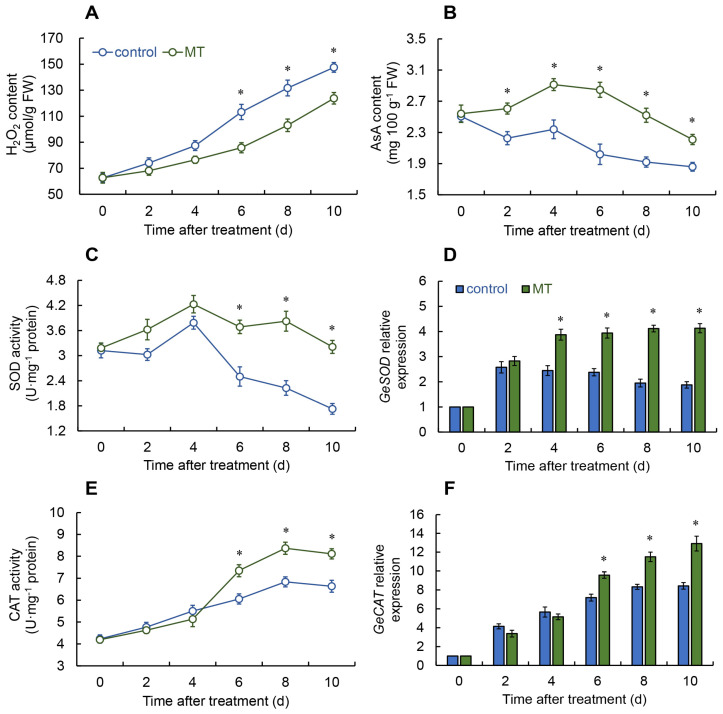
Changes in hydrogen peroxide (H_2_O_2_) (**A**) and ascorbic acid (AsA) content (**B**), and the activities and expressions of superoxide dismutase (SOD) (**C**,**D**), catalase (CAT) (**E**,**F**), and ascorbate peroxidase (APX) (**G**,**H**) in fresh *G. elata* after melatonin treatment during storage at room temperature. * denotes significant difference at the level of *p* < 0.05. Vertical bars represent the standard errors of the means (±SE).

**Figure 3 ijms-25-11752-f003:**
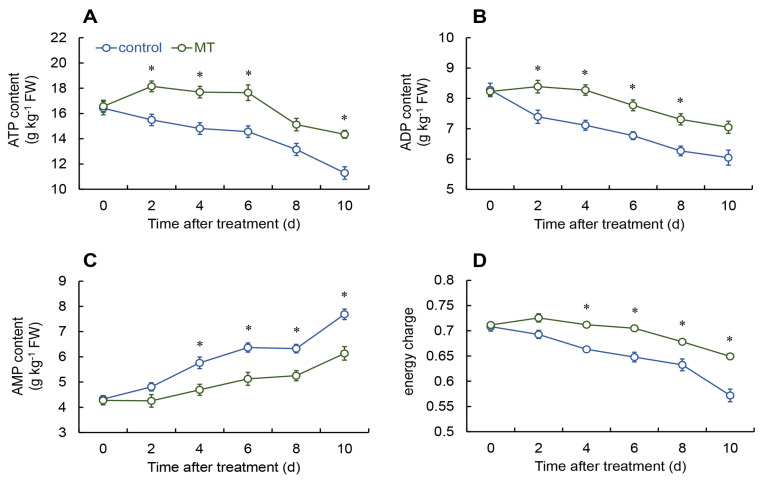
Changes in the content of adenosine triphosphate (ATP) (**A**), adenosine diphosphate (ADP) (**B**), adenosine monophosphate (AMP) (**C**), and energy charge (**D**) in fresh *G. elata* after melatonin treatment during storage at room temperature. * denotes significant difference at the level of *p* < 0.05. Vertical bars represent the standard errors of the means (±SE).

**Figure 4 ijms-25-11752-f004:**
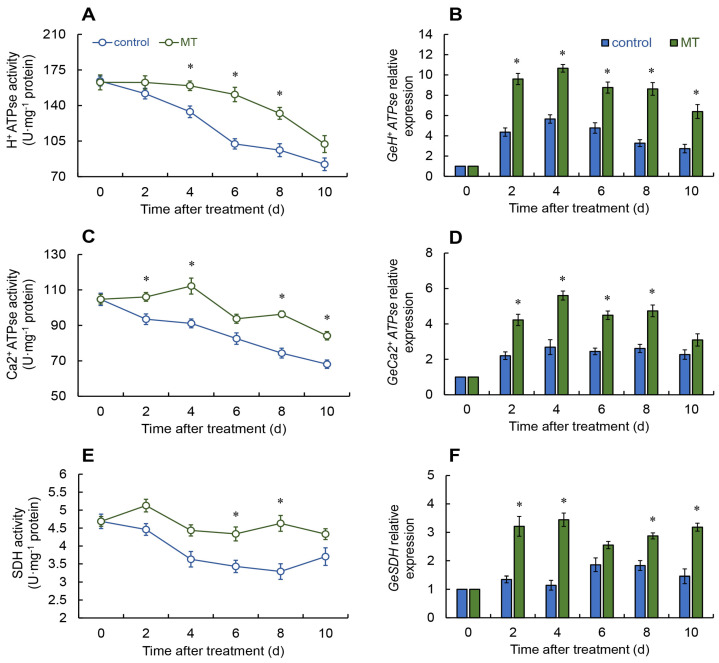
Changes in the activities and expressions of H^+^-ATPase (**A**,**B**), Ca2+-ATPse (**C**,**D**), cytochrome c oxidase (CCO) (**E**,**F**), and succinate dehydrogenase (SDH) (**G**,**H**) in fresh *G. elata* after melatonin treatment during storage at room temperature. * denotes significant difference at the level of *p* < 0.05. Vertical bars represent the standard errors of the means (±SE).

**Figure 5 ijms-25-11752-f005:**
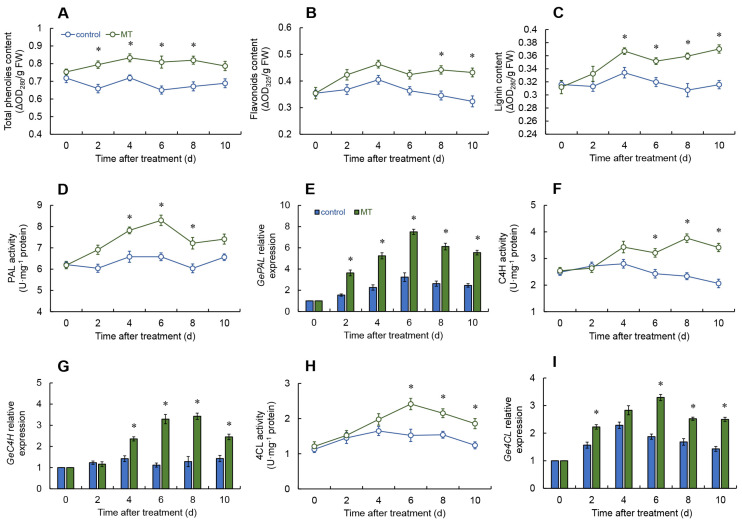
Changes in the content of total phenols (**A**), flavonoids (**B**), lignin (**C**) and the activities and expressions of phenylalanine ammonia-lyase (PAL) (**D**,**E**), cinnamate-4-hydroxylase (C4H) (**F**,**G**), and 4-coumarate-CoA ligase (4CL) (**H**,**I**) in fresh *G. elata* after melatonin treatment during storage at room temperature. * denotes significant difference at the level of *p* < 0.05. Vertical bars represent the standard errors of the means (±SE).

## Data Availability

The original contributions presented in the study are included in the article; further inquiries can be directed to the corresponding authors.

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
