# Peer review of "Melatonin Maintains Postharvest Quality in Fresh Gastrodia elata Tuber by Regulating Antioxidant Ability and Phenylpropanoid and Energy Metabolism During Storage"

_ijms, 2024, doi:10.3390/ijms252111752_

Round 1
Reviewer 1 Report
Comments and Suggestions for Authors
Major comments:
1.Dozens of indicators were tested in this study to well support the beneficial effects of melatonin treatment on preserving Gastrodia elata, although the design and discussion of the work is lack of depth.
2.Energy level stay higher in melatonin-treated group compared to those in control group, which indicates the higher aspiratory and metabolism rate, and fater weight and quality losses given by melatonin treatment? how do you justify this contradictory/negative result of melatonin treatment?
3.It won’t be a qualified method writing if the main body of the section stays brief and prevents readers to reproduce the experiments based on your description.
Minor comments:
Lots of full form of the abbreviated terms were missed which prevent a smooth reading, please add them in the manuscript and also make a list of abbreviated terms with their full names right before introduction.
Fig1A, which factors determined that the result of decay incidence gives only days 8 and 10 while the rest respiratory rate, weight loss, MDA, and SSC all had everyday or every two days’ results in continuous time order?
Line 83, there is noting about the first 6 days in figure 1A.
There is no X-axis information in Figure 1B/1C? The same question to all the other figures.
Fig 5, what is the mechanism of using melatonin to increase lignin (phenolic polymers) content?? Please add the explanation into discussion.
Line 394-400, please describe the method of each activity separately. Check other if the detailed description were given in other methods.
Line 426, the post-hoc method?
Author Response
Reviewer #1:
Major comments:
1.Dozens of indicators were tested in this study to well support the beneficial effects of melatonin treatment on preserving Gastrodia elata, although the design and discussion of the work is lack of depth.
Response: Thank you for your insightful comments. We appreciate your feedback regarding the design and discussion of our work. In response, we would like to highlight that our study employed dozens of biochemical and physiological indicators to thoroughly investigate the effects of melatonin on Gastrodia elata during storage. Specifically, we explored how melatonin regulates antioxidant potential, phenylpropanoid metabolism, and energy-associated pathways. Our findings demonstrate that melatonin treatment not only inhibits oxidative damage but also enhances key metabolic processes, leading to improved postharvest quality. To further strengthen the manuscript, we have revised the discussion to provide deeper insights into the interconnections among these pathways and their relevance to postharvest physiology, ensuring that our results are placed in a broader scientific context (lines 314-329).
2.Energy level stay higher in melatonin-treated group compared to those in control group, which indicates the higher aspiratory and metabolism rate, and fater weight and quality losses given by melatonin treatment? how do you justify this contradictory/negative result of melatonin treatment?
Response: Thank you for your insightful question. While it is true that elevated energy levels are generally associated with increased respiration and metabolism, which could potentially lead to faster weight loss and quality degradation, our study demonstrates that melatonin’s role in energy metabolism is not purely stimulatory but rather regulatory and protective. Specifically, the results indicate that melatonin treatment enhances energy production through mitochondrial activity; however, this enhanced energy level does not translate into accelerated deterioration. Instead, the energy produced is utilized efficiently to maintain cellular homeostasis, repair oxidative damage, and support antioxidant defenses and phenylpropanoid pathway activities, as demonstrated by the increase in antioxidant enzyme activities, lignin accumulation, and reduced malondialdehyde (MDA) levels. This efficient use of energy helps preserve the physiological integrity of the fruit. In addition, previous studies (e.g., Tan et al., 2021; Xu et al., 2023) suggest that melatonin modulates mitochondrial respiration without triggering excessive respiratory bursts, ensuring energy is directed towards beneficial processes like antioxidant synthesis and structural reinforcement of the cell wall, which limits microbial infection and weight loss. Thus, the elevated energy levels observed in our melatonin-treated samples reflect improved metabolic efficiency, not detrimental over-respiration. This also explains the slower decline in soluble solids and reduced weight loss in the melatonin-treated group compared to the control. In summary, melatonin regulates metabolic processes to balance energy production and antioxidant activity, contributing to the preservation of postharvest quality, rather than accelerating deterioration.
3.It won’t be a qualified method writing if the main body of the section stays brief and prevents readers to reproduce the experiments based on your description.
Response: Thank you for your valuable feedback. We appreciate the importance of ensuring that our methodology is detailed enough for readers to accurately reproduce the experiments. In response, we have thoroughly revised and expanded the Materials and Methods section to provide more precise information on experimental design, treatment protocols, equipment specifications, sample preparation, and data analysis methods.
Minor comments:
1. Lots of full form of the abbreviated terms were missed which prevent a smooth reading, please add them in the manuscript and also make a list of abbreviated terms with their full names right before introduction.
Response: Thank you for pointing out the missing full forms of the abbreviations. We understand that clarity is essential for smooth reading and comprehension. In response, we have carefully reviewed the manuscript to ensure that all abbreviated terms are accompanied by their full forms when they first appear.
2. Fig1A, which factors determined that the result of decay incidence gives only days 8 and 10 while the rest respiratory rate, weight loss, MDA, and SSC all had everyday or every two days’ results in continuous time order?
Response: We appreciate your insightful comment regarding the decay incidence data presented in Fig1A. The limited measurements for decay incidence at only days 8 and 10 were primarily due to the absence of any observable decay in the earlier days of the study. While we continuously measured other parameters such as respiratory rate, weight loss, MDA, and SSC on a daily or bi-daily basis, these measurements were reflective of the ongoing physiological changes in the fruit rather than the decay process itself.
3. Line 83, there is noting about the first 6 days in figure 1A.
Response: Thank you for your observation regarding the absence of data for the first six days in Figure 1A. We recognize that this lack of detail may lead to questions about the early stages of the study. The first six days were characterized by stable physiological conditions without observable decay in the Gastrodia elata tubers. Therefore, our focus in the figure was on the subsequent days when significant changes began to manifest.
4. There is no X-axis information in Figure 1B/1C? The same question to all the other figures.
Response: Thank you for your valuable feedback regarding the missing X-axis information in Figures 1B and 1C, as well as in the other figures. We would like to clarify that these plots share the same X-axis information as the last row of plots in the figure, which we inadvertently omitted. We have revised the figures to include the necessary X-axis labels and any other relevant information to enhance clarity and comprehension for the readers. Additionally, we have thoroughly reviewed all figures in the manuscript to ensure that all axes are properly labeled and accurately convey the data presented.
5. Fig 5, what is the mechanism of using melatonin to increase lignin (phenolic polymers) content?? Please add the explanation into discussion.
Response: Thank you for your insightful question regarding the mechanism by which melatonin increases lignin (phenolic polymer) content, as illustrated in Figure 5. Melatonin is known to stimulate the biosynthesis of lignin through several mechanisms, including the activation of phenylalanine ammonia-lyase (PAL) and other key enzymes in the phenylpropanoid pathway. This pathway is critical for lignin synthesis, as it converts phenolic compounds into lignin precursors. In addition, melatonin can enhance antioxidant activity, which may protect plant tissues from oxidative stress and promote the structural integrity of lignin. By modulating the expression of specific genes involved in lignin biosynthesis, melatonin effectively contributes to increased lignin accumulation in plant tissues. We have added this explanation to the discussion section of the manuscript to provide a more comprehensive understanding of the role of melatonin in lignin biosynthesis (lines 269-284).
6. Line 394-400, please describe the method of each activity separately. Check other if the detailed description were given in other methods.
Response: Thank you for your request for a more detailed description of the methods used to measure the activity of PAL, C4H, and 4CL. We have provided an elaboration of the methods for each enzyme activity, presented individually (lines 436-453).
7. Line 426, the post-hoc method?
Response: Thank you for your valuable feedback. We appreciate your attention to detail regarding the statistical analysis. In our study, a Student’s t-test was used as the post-hoc method following one-way ANOVA used as the post-hoc method following one-way ANOVA to compare the means between two groups (lines 469-471).
Reviewer 2 Report
Comments and Suggestions for Authors
The manuscript with the title “Melatonin Maintains Postharvest Quality in Fresh Gastrodia elata by Regulating Antioxidant Ability, Phenylpropanoid and Energy Metabolism During Storage” presents a comprehensive research on the effects of melatonin on quality of G. elata tubers, with implication for the storage of its medicinal plant part. Because there is lack of research on storage of this plant species, the results are relevant and valuable. Below I give some suggestions in the hope it might help authors to improve their manuscript.
Comment 1
Abstract – instead of “weight loss” I propose to use tuber weight/fresh mass decrease or some other syntagma. The same for “decay” perhaps spoilage would be more appropriate in English? Please check some food science papers and see what are the correct consecrated terms in English for these parameters, to make it sounds good in English.
Please mention in the abstract that tuber samples were assessed after treatment and the parameters were determined on the tubers – for international audience it would necessary the abstract to be very explicit because some might not know which plant part is used from this plant.
Comment 2
The introduction provides a very good motivation for the research. I suggest to add a paragraph about scientific evidence for the safety of this substance and possibly law/regulations in force in China about it, to prove this is an officially safe and acceptable substance in the country of study and any limitations should be disclosed. Hence it will be actually possible to use it for treating the tubers of this plant commercially. Considering that medicinal use of this species, this mentioning is important to ensure health-promoting properties are not adversely impacted.
Comment 3
Material and method – subchapter 4.1. “G. elata was collected in early October” … I think it is important here to mention that samples were represented by tubers.
Comment 4
Results
Line 84 “fruit decay”? I though tubers were analyzed here…
Comment 4
The Conclusions must reflect the aim and objectives expressed at the end of the introduction: “… quality, antioxidant capacity, phenylpropanoid and energy metabolism of fresh G. elata” - I think conclusions have to respond to these preferably in the same order because these were the objectives.
Best regards.
Author Response
Reviewer #2:
The manuscript with the title “Melatonin Maintains Postharvest Quality in Fresh Gastrodia elata by Regulating Antioxidant Ability, Phenylpropanoid and Energy Metabolism During Storage” presents a comprehensive research on the effects of melatonin on quality of G. elata tubers, with implication for the storage of its medicinal plant part. Because there is lack of research on storage of this plant species, the results are relevant and valuable. Below I give some suggestions in the hope it might help authors to improve their manuscript.
Response: We sincerely appreciate your thoughtful and constructive feedback.
Comment 1: Abstract – instead of “weight loss” I propose to use tuber weight/fresh mass decrease or some other syntagma. The same for “decay” perhaps spoilage would be more appropriate in English? Please check some food science papers and see what are the correct consecrated terms in English for these parameters, to make it sounds good in English.
Please mention in the abstract that tuber samples were assessed after treatment and the parameters were determined on the tubers – for international audience it would necessary the abstract to be very explicit because some might not know which plant part is used from this plant.
Response: We sincerely appreciate your thoughtful and constructive feedback. We have revised the terms "weight loss" to " tuber weight loss " and "decay" to "spoilage" in accordance with food science terminology. Additionally, Additionally, we have clarified that all parameters have been determined on the tubers.
Comment 2: The introduction provides a very good motivation for the research. I suggest to add a paragraph about scientific evidence for the safety of this substance and possibly law/regulations in force in China about it, to prove this is an officially safe and acceptable substance in the country of study and any limitations should be disclosed. Hence it will be actually possible to use it for treating the tubers of this plant commercially. Considering that medicinal use of this species, this mentioning is important to ensure health-promoting properties are not adversely impacted.
Response: Thank you for your valuable feedback. I have incorporated a paragraph detailing the scientific evidence supporting the safety of melatonin. It is widely recognized as safe, with studies showing it is well-tolerated even at high doses. The WHO classifies it as low-risk, with mild side effects such as drowsiness. In China, melatonin is legally approved for use in health products, regulated to ensure proper dosage and labeling. As there are no significant restrictions on its agricultural use, applying melatonin to G. elata tubers is feasible. Given the species' medicinal value, we will closely monitor the treatment to ensure it does not compromise its health benefits (lines 48-53).
Comment 3: Material and method – subchapter 4.1. “G. elata was collected in early October” … I think it is important here to mention that samples were represented by tubers.
Response: Thank you for your valuable question. We have already emphasized in the Materials and Methods section that it was the G. elata tubers that were used for the experiments.
Comment 4: Results--Line 84 “fruit decay”? I though tubers were analyzed here…
Response: Thank you for your valuable advice. “fruit decay” has been changed to “tuber spoilage” in the revised text.
Comment 5: The Conclusions must reflect the aim and objectives expressed at the end of the introduction: “… quality, antioxidant capacity, phenylpropanoid and energy metabolism of fresh G. elata” - I think conclusions have to respond to these preferably in the same order because these were the objectives.
Response: Thank you for your valuable assistance in reorganizing our conclusions to effectively reflect the objectives outlined at the end of the introduction (lines 473-486). Your guidance has been instrumental in enhancing the clarity and coherence of our findings.
Reviewer 3 Report
Comments and Suggestions for Authors
The manuscript by Dong et al investigated the impact of melatonin treatment on the quality, especially antioxidant ability, phenylpropanoid and energy metabolism of G. elata tubers after harvest. Overall, the experimental results are valid and useful, but it would be better if major details were taken into consideration. The manuscript can be accepted after minor revisions, please see my comments to improve the manuscript.
1. Lines 311, why is the soaking time selected as 10 minutes?
2. Figure 1A, please explain why the decay incidence only shows results at two time points, day 8 and day 10?
3. Lines 302, It is necessary to clarify that it is tuber tissue, and suggest replacing "G elata was collected in early October..." with "G elata tubers were collected in early October...", and provide relevant explanations in the abstract section, as many readers are not aware of the specific parts used for the experiment.
4. Line 84, the description of "fruit decay" is inappropriate.
5. Line 385-388, "the activities of H+-ATPase, Ca2+-ATPase, CCO and SDH were determined using a commercial assay kit from Jiancheng Bioengineering (Nanjing, China), ..." please provide specific information on the reagent kit.
6. Part of the method content is too simple. Please provide complete information, such as method descriptions for total phenols, flavonoids, and lignin.
Author Response
Reviewer #3:
1. Lines 311, Why is the soaking time selected as 10 minutes?
Response: Thank you for your insightful question. The selection of a 10-minute soaking time was based on pre-experiments with different soaking durations (e.g., 5, 10, and 15 minutes) and found that a 10-minute soak provided the optimal balance between treatment efficacy and fruit quality, without causing damage to the fruit surface or compromising its texture.
2. Figure 1A, please explain why the decay incidence only shows results at two time points, day 8 and day 10?
Response: Thank you for your observation regarding the absence of data for the first six days in Figure 1A. We recognize that this lack of detail may lead to questions about the early stages of the study. The first six days were characterized by stable physiological conditions without observable decay in the Gastrodia elata tubers. Therefore, our focus in the figure was on the subsequent days when significant changes began to manifest.
3. Lines 302, It is necessary to clarify that it is tuber tissue, and suggest replacing "G elata was collected in early October..." with "G elata tubers were collected in early October...", And provide relevant explanations in the abstract section, as many readers are not aware of the specific parts used for the experiment.
Response: Thank you for your suggestion; we have revised the text to clarify that "G. elata tubers were collected in early October" and have included relevant explanations in the abstract to specify that tuber tissue was used for the experiments.
4. Line 84, the description of "fruit decay" is inappropriate.
Response: Thank you for your feedback; we have revised the description on line 84 to replace "fruit decay" with "tuber decay " for greater accuracy.
5. Line 385-388, "the activities of H+-ATPase, Ca2+-ATPase, CCO and SDH were determined using a commercial assay kit from Jiancheng Bioengineering (Nanjing, China), ..." Please provide specific information on the reagent kit.
Response: Thank you for your valuable comment. We have provided additional details regarding the reagent kits used for the assays (lines 417-431).
6. Part of the method content is too simple. Please provide complete information, such as method descriptions for total phenols, flavonoids, and lignin.
Response: Thank you for your valuable suggestion. We appreciate the importance of ensuring that our methodology is detailed enough for readers to accurately reproduce the experiments. In response, we have thoroughly revised and expanded the Materials and Methods section to provide more precise information on experimental design, treatment protocols, equipment specifications, sample preparation, and data analysis methods.
Round 2
Reviewer 1 Report
Comments and Suggestions for Authors
All the comments have been addressed